# Cross-Attention Gradient Transplantation (CAGT): Mitigating Gradient Conflict in Multi-Task Deep Learning Through Cross-Blockwise Attention

## Abstract

Multi-task learning tries to use shared structure across tasks that are related by optimizing a joint objective. In practice though, gradients from individual tasks losses can cause conflict. This causes destructive interference that slows convergence and reduces performance. Previous approaches like PCGrad and CAGrad address this by projecting away or reweighting conflicting gradients at the vector level, but they do not explicitly exchange or reuse gradient components across tasks. This paper proposes Cross-Attention Gradient Transplantation (CAGT), which is a method that finds conflicting gradient components and replaces them using deterministic cosine similarity based cross-attention over gradient subblocks from other tasks. Task gradients are partitioned into subblocks at each optimization step, and for each subblock showing conflict, attention weights are computed from cosine similarities with other tasks' subblocks. The resulting weighted combination is rescaled to keep the original magnitude, and it is interpolated with the original subgradient. This produces adjusted task gradients that reduce adversarial interactions while keeping constructive signals. Experimental results show that CAGT outperforms traditional approaches such as PCGrad and CAGrad, achieving roughly 10% lower loss values on some multi task deep learning datasets, such as ROT-MNIST and CelebA.

## 1 Introduction

Cross-task deep learning seeks to train a single model on multiple related tasks simultaneously, using shared representations to improve data efficiency and generalization Caruana (1997). In principle, sharing parameters across tasks can help discover common structure and yield performance gains over training each task independently Baxter (2000). However, in practice, naively optimizing all tasks jointly usually leads to a difficult optimization problem. Jointly trained models sometimes perform worse on each task than separate single-task models Ruder (2017). This gap between theory and practice is partially because of the fact that the gradient signals from different tasks can interfere, which makes the multi-task loss landscape hard to navigate Yu et al. (2020).

A core source of difficulty is gradient conflict between tasks. We say two tasks have conflicting gradients when their update directions point away from each other (i.e. their cosine similarity is negative) Yu et al. (2020). These conflicts can stall learning, because one task's gradient update can partially cancel or even reverse another's; so, improving one task comes at the expense of another Liu et al. (2021a). Moreover, tasks usually produce gradients of very different magnitudes, so a large gradient can dominate the update even if it is misaligned with the other tasks' objectives Chen et al. (2018). Conflicting gradient directions and scale imbalances can prevent the optimizer from making progress on all tasks simultaneously Liu et al. (2021a); Chen et al. (2018).

In this work, we use cross-attention to directly modulate task gradients at a finer granularity than prior methods. We propose a framework called Cross-Attention Gradient Transplantation (CAGT), which learns to adaptively combine and reroute gradient signals across tasks at the level of gradient subspaces.

To better understand the benefits of finer grained gradient recomposition, we focus on three key aims: (1) Conflict reduction: quantify how cross block attention modulates task gradients to reduce negative interference and improve alignment between conflicting tasks; (2) Adaptive gradient combination: evaluate the effect of locally transplanting gradient components from one task to another, including the role of adaptive interpolation coefficients in controlling the strength of transplantation; (3) Empirical performance impact: measure whether these gradient modifications translate into improved task wise loss reduction and overall model convergence.

## 2  RELATED WORKS

Multi-task deep learning (MTL) has been extensively studied as a paradigm for improving generalization by leveraging shared representations across related tasks Caruana (1997); Baxter (2000). The fundamental premise is that jointly learning multiple tasks can lead to better performance than learning each task independently, especially when tasks share underlying structure or when data for individual tasks is limited Ruder (2017). However, the optimization challenges in MTL, like gradient conflicts between tasks, have motivated a rich body of research on conflict mitigation strategies. Further related works can be seen in Appendix A.

## 3  CROSS-ATTENTION GRADIENT TRANSPLANTATION

In multi-task deep learning, a model must optimize multiple task-specific losses simultaneously. Directly summing gradients from all tasks can create conflicts, where updates from some tasks interfere with others. *Cross-Attention Gradient Transplantation* (CAGT) is a method to detect and resolve these conflicts at a fine-grained level before updating model parameters.

CAGT first partitions model parameters into fixed blocks, treating the gradient within each block as an atomic unit. For each task and block, it measures how well the task's gradient aligns with gradients from other tasks. Blocks that are negatively aligned, indicating potential conflict, are flagged for adjustment, while well aligned blocks remain unchanged.

For conflicting blocks, CAGT reconstructs a replacement gradient using a form of cross-task attention: each task's blocked gradient is compared with others, and an attention-weighted combination of the other tasks' gradients is computed. This produces a direction that is more compatible with the multi-task objective. The reconstructed gradient is then scaled to match the original magnitude and interpolated with the original block, allowing smooth adjustment rather than abrupt replacement.

Once all blocks are processed, the modified gradients are reassembled into a full task gradient. These adjusted gradients are then aggregated across tasks to produce the final update applied by any standard gradient-based optimizer.

**Implementation notes.**   Gradients are flattened and divided into blocks. Conflicts are detected via block-level similarity metrics, and cross-task attention is applied only to conflicting blocks. The interpolation factor and attention mechanism are flexible, allowing adaptation to different architectures and tasks.

The full algorithmic formulation, including the exact attention computation and strategies for adaptive interpolation, is provided in Appendix Sections B and C. Theoretical proofs and guarantees can be found in Appendix D.

## 4  METHOD

### 4.1  EXPERIMENTAL SETUP

Experiments are conducted on two multi-task learning benchmarks.

Rotated MNIST is derived from the MNIST dataset by augmenting each image with a random rotation selected from $\{0°, 90°, 180°, 270°\}$ and a randomly assigned color palette from six predefined options. The benchmark consists of three tasks: digit classification, rotation classification, and color classification. The dataset contains 60,000 training images and 10,000 test images.

CelebA is used for multi-task facial attribute prediction. It focuses on five binary attributes: Smiling, Male, Eyeglasses, WearingHat, and Bangs. We follow the standard training and validation split, consisting of 162,770 training images and 19,867 validation images.

For Rotated MNIST, we use a convolutional neural network backbone with a shared feature extractor and task-specific output heads. For CelebA, we use a ResNet-18 backbone pretrained on ImageNet as the shared encoder. Five parallel binary classification heads are attached to the final feature representation (one for each attribute). All models are trained using the AdamW optimizer with a learning rate of $2 \times 10^{-3}$, weight decay of $10^{-4}$, and batch size of 256. Cross-entropy loss is used for all tasks. For Rotated MNIST, task weights of 1.0, 0.8, and 0.8 are applied to the digit, rotation, and color tasks, respectively. For CelebA, all tasks are weighted equally.

## 5 RESULTS

We present comprehensive experimental results evaluating Cross-Attention Gradient Transplantation (CAGT) against SOTA multi-task optimization methods. Our experiments show that CAGT consistently reduces gradient conflicts and improves task performance across diverse multi-task learning scenarios.

### 5.1 MAIN EXPERIMENTAL RESULTS

#### 5.1.1 ROTATED MNIST MULTI-TASK LEARNING

Table 1 shows results on the Rot-MNIST benchmark, which presents a multi-task scenario with three diverse tasks (digit classification, rotation prediction, and color classification). CAGT demonstrates better performance, as it achieved the lowest average validation loss (0.2525) and highest accuracy across all three tasks. Data was averaged over experimental runs conducted with 3 varying seeds and 3 reruns per seed.

Table 1: Comparison of CAGT (ours), PCGrad, GradNorm, MGDA, and CAGrad on Rot-MNIST (Digit, Rotation, and Color classification). Average validation losses are shown over 3 seeds with 3 reruns, full table can be found in Appendix E

| Method | Loss ($\downarrow$) | Dig. Acc. ($\uparrow$) | Rot. Acc. ($\uparrow$) | Col. Acc. ($\uparrow$) |
|---|---|---|---|---|
| PCGrad | 0.2957 | 92.14% | 97.98% | 99.96% |
| GradNorm | 0.3647 | 92.53% | 96.57% | 99.83% |
| MGDA | 0.3027 | 92.12% | 98.02% | **99.99%** |
| CAGrad | 0.2797 | 91.96% | 97.44% | **99.99%** |
| CAGT (Ours) | **0.2525** | **93.57%** | **98.11%** | **99.99%** |

#### 5.1.2 CELEBA ATTRIBUTE CLASSIFICATION

Table 2 compares different multi-task optimization methods on CelebA. CAGT consistently outperforms the other methods, achieving the highest average AUROC ($0.9874 \pm 0.0063$) and the lowest loss ($0.1598 \pm 0.014$), indicating both more accurate predictions and more confident outputs.

Table 2: Comparison of multi-task optimization methods on CelebA. Tasks: Smiling, Male, Eyeglasses, Wearing Hat, Bangs. Trained for 20 epochs.

| Method | Avg. AUROC ($\uparrow$) | Avg. Loss ($\downarrow$) |
|---|---|---|
| GradNorm | $0.9832 \pm 0.0024$ | $0.1987 \pm 0.010$ |
| MGDA | $0.9845 \pm 0.0042$ | $0.1934 \pm 0.010$ |
| PCGrad | $0.9851 \pm 0.0037$ | $0.1859 \pm 0.007$ |
| CAGrad | $0.9863 \pm 0.0021$ | $0.1728 \pm 0.008$ |
| CAGT (Ours) | $\mathbf{0.9874 \pm 0.0063}$ | $\mathbf{0.1598 \pm 0.014}$ |

## 5.2 ABLATION STUDIES

We conduct ablation studies to assess the sensitivity of CAGT to its primary hyperparameters and to isolate the contribution of key design choices. In particular, we analyze (i) the interaction between attention temperature $\tau$ and gradient interpolation coefficient $\lambda$, which jointly control attention sharpness and the strength of gradient transplantation, and (ii) the choice of attention similarity mechanism. All ablations are performed on ROT-MNIST using an identical architecture and training protocol, and trained for 7 epochs with identical batch sizes. Validation loss is used as the primary model selection criterion. The collected data is an average across 3 runs per seed, over 3 seeds. The results and analysis taken from the ablation studies can be seen in Appendix E.1.

## 6 DISCUSSION

Our experiments show that Cross-Attention Gradient Transplantation (CAGT) effectively addresses gradient conflicts in multi-task learning through a novel cross-attention based mechanism.

### 6.1 KEY FINDINGS

Our results support several key observations. First, gradient conflicts in deep learning MTL scenarios are highly localized rather than uniformly distributed across parameters. Partitioning gradients into parameter blocks allows CAGT to find and resolve conflicts at a finer granularity than global methods such as PCGrad or CAGrad, which operate on entire gradients.

Second, cross-task gradient information can be beneficial even in the presence of conflict. The attention patterns show that gradients from non-conflicting tasks can provide useful descent directions for conflicting subspaces. This finding contrasts with projection based approaches that suppress or discard conflicting components.

Third, adaptive conflict resolution is useful for effective optimization. Early stage training favors stronger gradient transplantation when conflicts are pronounced, while later stages increasingly preserve original gradients as task alignment improves. This means that CAGT could reasonably be used only for early epochs, and be reverted back to normal learning for further epochs in order to balance performance with validation gains.

Finally, preserving gradient magnitude contributes to training stability. By rescaling reconstructed gradients to match the original norm, CAGT avoids abrupt changes in optimization dynamics that can potentially destabilize training.

### 6.2 LIMITATIONS

CAGT introduces additional computational and memory overhead due to gradient partitioning and attention computation. The method is sensitive to hyperparameters such as block size, temperature, and interpolation coefficient, which requires tuning across architectures via hyperparameter sweeps. Scalability is limited by the quadratic dependence on the number of tasks, and a complete theoretical characterization in non-convex settings remains an open problem.

### 6.3 FUTURE DIRECTIONS

Future work includes hierarchical or task-aware gradient partitioning, improved scalability for large task sets, tighter theoretical analysis of learned attention dynamics, and extensions to continual and multi-modal learning scenarios. Also, systems where CAGT deactivates after a learnable $N$ number of epochs in order to balance performance with reasonable gains, is a potential direction to pursue.

## 7 CONCLUSION

This work introduces Cross-Attention Gradient Transplantation (CAGT). By partitioning gradients into subspaces and using attention, CAGT selectively reconstructs conflicting gradients while preserving complementary information. Empirical results on Rot-MNIST and CelebA demonstrate

that CAGT consistently outperforms existing approaches such as PCGrad, CAGrad, MGDA, and GradNorm, and reduces loss and improves task-specific accuracies.

Future work will explore improved scalability to larger task sets, hierarchical or task-aware gradient partitioning, and applications in continual learning scenarios. Overall, the empirical results show consistent improvements across benchmarks, which indicates that subblock and attention based gradient modification is a promising direction for multi-task optimization.

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

## A  FURTHER RELATED WORKS

Multi-task learning (MTL) has been extensively studied as a paradigm for improving generalization by leveraging shared representations across related tasks Caruana (1997); Baxter (2000). The fundamental premise is that jointly learning multiple tasks can lead to better performance than learning each task independently, especially when tasks share underlying structure or when data for individual tasks is limited Ruder (2017). However, the optimization challenges in MTL, like gradient conflicts between tasks, have motivated a rich body of research on conflict mitigation strategies.

### A.1  GRADIENT CONFLICT MITIGATION METHODS

Early work in multi-task optimization focused on static approaches like as task weighting and loss balancing. Cipolla et al. Cipolla et al. (2018) proposed uncertainty-based weighting that automatically balances tasks based on their homoscedastic uncertainty. Chen et al. Chen et al. (2018) introduced GradNorm, which dynamically adjusts task weights to normalize gradient magnitudes across tasks. These methods address some scale imbalances, but they do not directly handle directional conflicts between task gradients.

More recent approaches explicitly address gradient conflicts through geometric operations. Gradient Surgery (PCGrad) Yu et al. (2020) projects conflicting gradient components onto the orthogonal space of other tasks, effectively removing negative interference. CAGrad Liu et al. (2021a) takes a different approach by finding a descent direction that maximizes the minimum improvement across all tasks through a constrained optimization formulation. Multiple Gradient Descent Algorithm (MGDA) Sener and Koltun (2018) treats multi-task optimization as a multi-objective problem and finds Pareto-optimal solutions in the span of task gradients. Recent work has also explored Impartial Multi-Task Learning (IMTL), which aims to ensure that no single task dominates the optimization process by balancing gradient contributions across tasks Liu et al. (2021b).

Another approach is Nash-MTL Navon et al. (2022), which frames multi-task gradient combination as a bargaining game, arriving at the Nash Bargaining Solution as a principled joint update direction. Unlike these projection or optimization based methods, our work introduces a fundamentally different mechanism through attention based gradient transplantation. Other conflict mitigation strategies include stochastic approaches like Gradient Sign Dropout Chen et al. (2020), which randomly drops gradient signs to reduce interference.

Our work builds upon these conflict aware methods but introduces a fundamentally different mechanism. Instead of projecting away or reweighting conflicting components, we actively transplant beneficial gradient information across tasks using learned attention mechanisms.

### A.2  ATTENTION MECHANISMS IN MULTI-TASK LEARNING

Attention mechanisms have revolutionized representation learning across domains from natural language processing Vaswani et al. (2017) to computer vision Dosovitskiy et al. (2021). In multi-task settings, attention has been primarily used for feature-level information sharing rather than gradient-level optimization. Cross-Stitch Networks Misra et al. (2016), which learns soft attention masks to combine feature maps from task-specific networks, enable adaptive feature sharing. Similarly, Multi-Task Attention Networks (MTAN) Liu et al. (2019) extend this idea with soft-attention modules over shared backbone features to generate task-specific representations.

These methods show the effectiveness of attention for feature-level information sharing but they operate in the forward pass and do not address optimization challenges in the backward pass. Our work applies attention mechanisms directly to gradient signals for conflict mitigation.

### A.3  GRADIENT-BASED META-LEARNING AND OPTIMIZATION

The idea of modifying gradient updates based on learned criteria has connections to gradient-based meta-learning approaches. Methods such as MAML Finn et al. (2017) learn initialization parameters that allow rapid adaptation to new tasks through gradient descent. Other approaches learn to modify gradients directly, such as gradient clipping Pascanu et al. (2013) and gradient regularization techniques Novak et al. (2018).

Recent work in learned optimization has explored using neural networks to predict update directions Andrychowicz et al. (2016); Li and Malik (2018). While these methods learn optimizers from scratch, our approach is more constrained and interpretable: we use attention mechanisms within a well defined theoretical framework to specifically address multi-task gradient conflicts.

### A.4 SUBSPACE AND BLOCK-WISE OPTIMIZATION

The partitioning of gradients into subspaces in our method relates to work on block-wise and subspace optimization techniques. Block coordinate descent methods Richtárik and Takáč (2016) optimize parameters in blocks rather than jointly, which can improve efficiency and convergence.

In the context of multi-task learning, modular and sparse approaches Mallya and Lazebnik (2017); Rosenbaum et al. (2017) have been explored to reduce interference by allocating different parameter subsets to different tasks. Our gradient partitioning approach is complementary. These methods work at the parameter level, but CAGT works at the gradient level, allowing it to maintain a fully shared model while still addressing local conflicts.

Our work combines ideas from these diverse areas. We combine the geometric insights of gradient conflict methods with the adaptive information sharing of attention mechanisms while operating at a subspace level.

## B  CROSS-ATTENTION GRADIENT TRANSPLANTATION

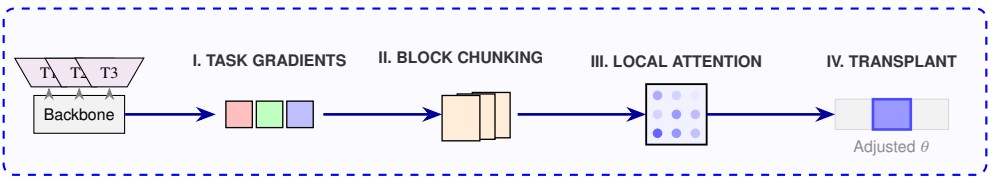

We consider a standard multi-task learning setting in which a single model with parameters $\theta \in \mathbb{R}^d$ is trained to minimize $T$ task-specific losses $\{\mathcal{L}_i\}_{i=1}^{T}$. At each optimization step, the model receives a collection of task gradients

$$\mathbf{g}_i = \nabla_\theta \mathcal{L}_i(\theta), \quad i = 1, \ldots, T.$$

Cross-Attention Gradient Transplantation (CAGT) defines a deterministic transformation that maps the set $\{\mathbf{g}_i\}_{i=1}^{T}$ to a new set of adjusted gradients $\{\mathbf{g}_i^{\text{CAGT}}\}_{i=1}^{T}$, which are then aggregated and passed to a standard gradient-based optimizer.

### B.1 GRADIENT PARTITIONING

Let $\theta$ be partitioned into $K$ disjoint parameter groups,

$$\theta = [\theta^{(1)}, \theta^{(2)}, \ldots, \theta^{(K)}],$$

where each $\theta^{(k)} \in \mathbb{R}^{d_k}$ and $\sum_{k=1}^{K} d_k = d$. This induces a corresponding decomposition of each task gradient,

$$\mathbf{g}_i = [\mathbf{g}_i^{(1)}, \mathbf{g}_i^{(2)}, \ldots, \mathbf{g}_i^{(K)}], \quad \mathbf{g}_i^{(k)} = \nabla_{\theta^{(k)}} \mathcal{L}_i(\theta).$$

The partitioning scheme is fixed throughout training. Each sub-gradient $\mathbf{g}_i^{(k)}$ is treated as an atomic unit for conflict assessment and transformation.

### B.2 LOCAL GRADIENT AGREEMENT MEASURE

For each task $i$ and sub-block $k$, CAGT computes

$$s_i^{(k)} = \frac{1}{T-1} \sum_{j \neq i} \frac{\mathbf{g}_i^{(k)} \cdot \mathbf{g}_j^{(k)}}{\|\mathbf{g}_i^{(k)}\| \, \|\mathbf{g}_j^{(k)}\|},$$

measuring the average cosine similarity between task $i$ and the remaining tasks within subspace $k$. Sub-blocks with $s_i^{(k)} < 0$ are designated as *conflicting*, while sub-blocks with $s_i^{(k)} \geq 0$ are preserved without modification.

### B.3 ATTENTION-BASED GRADIENT RECONSTRUCTION

For each conflicting $(i, k)$, CAGT constructs a replacement direction by cross attending over corresponding sub-gradients from other tasks. In the implemented formulation, attention is computed directly in normalized gradient space without learned projections, in order to improve performance. Let

$$\mathcal{J}_i = \{j \in \{1, \ldots, T\} \mid j \neq i\}.$$

We compute temperature-scaled attention weights

$$\alpha_{ij}^{(k)} = \frac{\exp\left(\frac{\mathbf{g}_i^{(k)} \cdot \mathbf{g}_j^{(k)}}{\tau \|\mathbf{g}_i^{(k)}\| \|\mathbf{g}_j^{(k)}\|}\right)}{\sum_{l \in \mathcal{J}_i} \exp\left(\frac{\mathbf{g}_i^{(k)} \cdot \mathbf{g}_l^{(k)}}{\tau \|\mathbf{g}_i^{(k)}\| \|\mathbf{g}_l^{(k)}\|}\right)},$$

and construct an attention-weighted recomposition

$$\tilde{\mathbf{g}}_i^{(k)} = \sum_{j \in \mathcal{J}_i} \alpha_{ij}^{(k)} \mathbf{g}_j^{(k)}.$$

### B.4 MAGNITUDE PRESERVATION AND INTERPOLATION

We rescale the reconstructed gradient to match the original magnitude

$$\hat{\mathbf{g}}_i^{(k)} = \frac{\|\mathbf{g}_i^{(k)}\|}{\|\tilde{\mathbf{g}}_i^{(k)}\| + \epsilon} \tilde{\mathbf{g}}_i^{(k)},$$

and define the transplanted gradient as

$$\mathbf{g}_i^{(k)\,\text{CAGT}} = (1 - \lambda)\mathbf{g}_i^{(k)} + \lambda\hat{\mathbf{g}}_i^{(k)},$$

with $\lambda \in [0, 1]$. Non-conflicting blocks are left unchanged.

### B.5 GRADIENT ASSEMBLY AND OPTIMIZATION

The adjusted task gradient is reconstructed by concatenation

$$\mathbf{g}_i^{\text{CAGT}} = [\mathbf{g}_i^{(1)\,\text{CAGT}}, \ldots, \mathbf{g}_i^{(K)\,\text{CAGT}}],$$

and the final update direction is obtained by

$$\Delta\theta = \sum_{i=1}^{T} \mathbf{g}_i^{\text{CAGT}}.$$

**Implementation details.** Gradients are flattened and partitioned into fixed-size blocks over parameters shared by all tasks. Task similarity is computed per block using normalized gradients, and blocks with average similarity below the threshold $\delta$ are marked as conflicting. Cross-task attention is then used to reconstruct gradients for conflicting tasks only, followed by interpolation with the original gradients. The specific attention mechanism and the algorithm for adaptive $\lambda$ are modular and deferred to Appendix C and Appendix C.2, respectively.

### B.6

CAGT Algorithm

---

**Algorithm 1** Cross-Attention Gradient Transplantation (CAGT)

---

**Require:** Task objectives $\{\mathcal{L}_i\}_{i=1}^{T}$, block size $B$, temperature $\tau$, base interpolation coefficient $\lambda$, conflict threshold $\delta$

1: Compute task gradients $\{\mathbf{g}_i\}_{i=1}^{T}$ and pack into flat vectors
2: Identify shared parameters and partition gradients into $K$ blocks of size $B$
3: Normalize gradients within each block
4: **for** $k = 1$ to $K$ **do**
5:     Compute task similarity scores $\{s_i^{(k)}\}$ over shared parameters
6:     Identify conflicting tasks $\mathcal{C}^{(k)} = \{i \mid s_i^{(k)} < \delta\}$
7:     **for** $i \in \mathcal{C}^{(k)}$ **do**
8:         Compute cross-task attention weights $\{\alpha_{ij}^{(k)}\}$
9:         $\tilde{\mathbf{g}}_i^{(k)} \leftarrow \sum_j \alpha_{ij}^{(k)} \mathbf{g}_j^{(k)}$
10:        Interpolate $\mathbf{g}_i^{(k)} \leftarrow (1 - \lambda_i^{(k)})\mathbf{g}_i^{(k)} + \lambda_i^{(k)}\tilde{\mathbf{g}}_i^{(k)}$
11:     **end for**
12: **end for**
13: Merge blocks, sum across tasks, and apply update $\Delta\theta$
14: **return** $\Delta\theta$

---

## C   Attention Mechanism Options and Adaptive Interpolation

CAGT supports multiple attention mechanisms and adaptive interpolation, which improve gradient reconstruction and conflict resolution across tasks. These options extend the basic deterministic cosine attention described in Section D.0.4. Algorithmic descriptions and mathematical formulations are provided below.

### C.1   Attention Mechanism Variants

CAGT computes cross-task attention within each block using either cosine, cosine squared, or euclidean attention. Cosine similarity is the standard attention using normalized gradients. Cosine squared similarity sharpens attention by squaring the magnitude of the cosine similarity while preserving the sign. Negative euclidean distance assigns higher attention to gradients that are closer in Euclidean space

---

**Algorithm 2** Compute CAGT Attention Weights

---

**Require:** Normalized task gradients $G \in \mathbb{R}^{T \times K \times B}$, attention mode, temperature $\tau$

1: **for** $k = 1$ to $K$ **do**
2:     Compute pairwise scores $S^{(k)}$ according to mode:
3:     **if** mode = 'cosine' **then**
4:         $S_{ij}^{(k)} \leftarrow G_i^{(k)} \cdot G_j^{(k)}$
5:     **else if** mode = 'cosine_squared' **then**
6:         $c \leftarrow G_i^{(k)} \cdot G_j^{(k)}$
7:         $S_{ij}^{(k)} \leftarrow c \cdot |c|$
8:     **else if** mode = 'euclidean' **then**
9:         $S_{ij}^{(k)} \leftarrow -\|G_i^{(k)} - G_j^{(k)}\|$
10:     **end if**
11:     Mask self-attention: $S_{ii}^{(k)} \leftarrow -\infty$
12:     Apply temperature scaling: $S^{(k)} \leftarrow S^{(k)}/\tau$
13:     Compute attention weights: $\alpha^{(k)} \leftarrow \mathrm{softmax}(S^{(k)})$
14: **end for**
15: **return** Attention weights $\alpha \in \mathbb{R}^{K \times T \times T}$

---

## C.2  ADAPTIVE INTERPOLATION COEFFICIENT

CAGT optionally adapts the interpolation coefficient $\lambda$ per step based on conflict severity and training progress. This allows stronger transplantation when conflicts are frequent and early in training, and weaker updates in later stages.

---

**Algorithm 3** Compute Adaptive $\lambda$ for CAGT

---

**Require:** Base $\lambda_{\text{base}}$, conflict ratio $r_{\text{conflict}}$, iteration $t$
 1: **if** adaptive $\lambda$ disabled **then**
 2:    $\lambda \leftarrow \lambda_{\text{base}}$
 3: **else if** $t < 500$ **then**
 4:    $\lambda \leftarrow \min(1.5\lambda_{\text{base}}, 0.5)$
 5: **else if** $t < 2000$ **then**
 6:    $\lambda \leftarrow \lambda_{\text{base}}$
 7: **else**
 8:    $\lambda \leftarrow \lambda_{\text{base}} \cdot (1 + 0.5 r_{\text{conflict}}) \cdot \max(0.5, 1 - (t - 2000)/10000)$
 9: **end if**
10: **return** $\lambda$

---

# D  THEORETICAL ANALYSIS AND PROOFS

We provide a theoretical justification for Cross-Attention Gradient Transplantation (CAGT) under a simplified convex setting. Our analysis explicitly accounts for (i) block-local operation, (ii) deterministic cosine-based attention, and (iii) magnitude-preserving interpolation. For clarity, we consider a single parameter block and two tasks.

### D.0.1  PROBLEM SETUP

**Definition D.1** (Two-Task Single-Block Setting). Let $\mathcal{L}_1, \mathcal{L}_2 : \mathbb{R}^d \to \mathbb{R}$ be convex and differentiable. Define

$$\mathcal{L}(\theta) = \mathcal{L}_1(\theta) + \mathcal{L}_2(\theta),$$

with block-restricted gradients $\mathbf{g}_i = \nabla \mathcal{L}_i(\theta)$. Assume $\mathcal{L}$ is $L$-smooth:

$$\|\nabla \mathcal{L}(\theta) - \nabla \mathcal{L}(\theta')\| \leq L\|\theta - \theta'\|.$$

We focus on the conflicting regime $\mathbf{g}_1^\top \mathbf{g}_2 < 0$.

### D.0.2  CAGT BLOCK UPDATE

Define the magnitude-rescaling operator

$$\mathcal{R}(\mathbf{a}, \mathbf{b}) := \frac{\|\mathbf{a}\|}{\|\mathbf{b}\|}\mathbf{b}.$$

For two tasks, deterministic attention reduces to mutual exchange:

$$\mathbf{g}_1^{\text{CAGT}} = (1 - \lambda)\mathbf{g}_1 + \lambda\,\mathcal{R}(\mathbf{g}_1, \mathbf{g}_2), \tag{1}$$
$$\mathbf{g}_2^{\text{CAGT}} = (1 - \lambda)\mathbf{g}_2 + \lambda\,\mathcal{R}(\mathbf{g}_2, \mathbf{g}_1). \tag{2}$$

The block update is

$$\mathbf{g}^{\text{CAGT}} = \mathbf{g}_1^{\text{CAGT}} + \mathbf{g}_2^{\text{CAGT}}.$$

### D.0.3  KEY PROPERTIES

- **Descent guarantee (Theorem D.2, Appendix D.0.4):** $\mathbf{g}^{\text{CAGT}}$ is a descent direction for $\mathcal{L}$ for any $\lambda \in [0, 1)$.

- **Conflict reduction (Theorem D.3, Appendix D.0.4)**: Cosine similarity between transformed gradients strictly increases.

- **Magnitude preservation (Lemma D.4, Appendix D.0.4)**: $\|\mathbf{g}_i^{\text{CAGT}}\| = \|\mathbf{g}_i\|$.

- **Single-step improvement (Theorem D.5, Appendix D.0.4)**: If $(\mathbf{g}_1 + \mathbf{g}_2)^\top \mathbf{g}^{\text{CAGT}} > \|\mathbf{g}_1 + \mathbf{g}_2\|^2$, a CAGT step decreases the joint loss more than naive summation.

### D.0.4 REMARKS

- The analysis abstracts deterministic cosine attention as a scalar weight which is exact for $T = 2$ and preserves the main geometric properties of the full algorithm.

- For multiple blocks, the CAGT update sums independent block updates and preserves descent guarantees by linearity.

- For $T > 2$, attention is distributed over several tasks.

Deterministic cross-block cosine attention with magnitude-preserving interpolation produces a valid descent update in the presence of gradient conflict. The analysis shows that CAGT not only maintains descent guarantees but also actively reduces inter-task gradient conflict.

## D.1 SCALAR QUANTITIES AND RESCALING

Define
$$A = \|\mathbf{g}_1\|^2, \quad B = \|\mathbf{g}_2\|^2, \quad C = \mathbf{g}_1^\top \mathbf{g}_2 < 0, \quad r = \sqrt{A/B}.$$

Then
$$\mathcal{R}(\mathbf{g}_1, \mathbf{g}_2)^\top (\mathbf{g}_1 + \mathbf{g}_2) = r(B + C), \tag{3}$$
$$\mathcal{R}(\mathbf{g}_2, \mathbf{g}_1)^\top (\mathbf{g}_1 + \mathbf{g}_2) = r^{-1}(A + C). \tag{4}$$

## D.2 DESCENT DIRECTION GUARANTEE (THEOREM D.2)

**Theorem D.2** (Block-Local CAGT Descent). *For any $\lambda \in [0, 1)$, $\mathbf{g}^{CAGT}$ is a descent direction for $\mathcal{L}$.*

*Proof.* From equation 1–equation 2,
$$\mathbf{g}^{\text{CAGT}} = (1 - \lambda)(\mathbf{g}_1 + \mathbf{g}_2) + \lambda(\mathcal{R}(\mathbf{g}_1, \mathbf{g}_2) + \mathcal{R}(\mathbf{g}_2, \mathbf{g}_1)). \tag{5}$$

Taking inner product with $\mathbf{g}_1 + \mathbf{g}_2$ and using the scalar relations above:
$$(\mathbf{g}_1 + \mathbf{g}_2)^\top \mathbf{g}^{\text{CAGT}} = (1 - \lambda)(A + B + 2C) + \lambda \left[ r(B + C) + r^{-1}(A + C) \right]. \tag{6}$$

Rewriting,
$$= A + B + 2C + \lambda \left[ rB + A/r + C(r + 1/r) - (A + B + 2C) \right].$$

By AM–GM, $r + 1/r \geq 2$ and $rB + A/r \geq A + B$. Since $C < 0$, the bracketed term is nonnegative. Hence,
$$(\mathbf{g}_1 + \mathbf{g}_2)^\top \mathbf{g}^{\text{CAGT}} \geq \|\mathbf{g}_1 + \mathbf{g}_2\|^2 > 0.$$
$\square$

## D.3 GRADIENT ALIGNMENT IMPROVEMENT (THEOREM D.3)

**Theorem D.3** (Conflict Reduction). *For any $\lambda > 0$,*
$$\cos(\mathbf{g}_1^{CAGT}, \mathbf{g}_2^{CAGT}) > \cos(\mathbf{g}_1, \mathbf{g}_2).$$

*Proof.* Write the transformed gradients in terms of the rescaled rotations:
$$\mathbf{g}_1^{\text{CAGT}} = (1 - \lambda)\mathbf{g}_1 + \lambda\mathcal{R}(\mathbf{g}_1, \mathbf{g}_2), \quad \mathbf{g}_2^{\text{CAGT}} = (1 - \lambda)\mathbf{g}_2 + \lambda\mathcal{R}(\mathbf{g}_2, \mathbf{g}_1).$$

By Lemma D.4, $\|\mathbf{g}_i^{\mathrm{CAGT}}\| = \|\mathbf{g}_i\|$, so the cosine similarity reduces to

$$\cos(\mathbf{g}_1^{\mathrm{CAGT}}, \mathbf{g}_2^{\mathrm{CAGT}}) = \frac{(\mathbf{g}_1^{\mathrm{CAGT}})^\top (\mathbf{g}_2^{\mathrm{CAGT}})}{\|\mathbf{g}_1\|\|\mathbf{g}_2\|}.$$

Expand the numerator:

$$(\mathbf{g}_1^{\mathrm{CAGT}})^\top (\mathbf{g}_2^{\mathrm{CAGT}}) = (1-\lambda)^2 \mathbf{g}_1^\top \mathbf{g}_2 + \lambda(1-\lambda)\big[\mathbf{g}_1^\top \mathcal{R}(\mathbf{g}_2, \mathbf{g}_1) + \mathbf{g}_2^\top \mathcal{R}(\mathbf{g}_1, \mathbf{g}_2)\big] + \lambda^2 \mathcal{R}(\mathbf{g}_1, \mathbf{g}_2)^\top \mathcal{R}(\mathbf{g}_2, \mathbf{g}_1)$$
$$= C(1-\lambda)^2 + \lambda(1-\lambda)\big[r^{-1}(A+C) + r(B+C)\big] + \lambda^2(\mathbf{g}_1^\top \mathbf{g}_2 \text{ rotated term}).$$

Using AM–GM as in Theorem D.2, the cross terms satisfy

$$r^{-1}A + rB \geq A + B, \quad r + r^{-1} \geq 2, \quad \text{and } C < 0,$$

so the numerator is strictly larger than $C = \mathbf{g}_1^\top \mathbf{g}_2$. Dividing by the unchanged norms then gives

$$\cos(\mathbf{g}_1^{\mathrm{CAGT}}, \mathbf{g}_2^{\mathrm{CAGT}}) > \cos(\mathbf{g}_1, \mathbf{g}_2).$$

Hence, the CAGT transformation improves alignment. $\qquad\square$

## D.4 Magnitude Preservation (Lemma D.4)

**Lemma D.4** (Norm Preservation). *For all $\lambda \in [0, 1]$,*

$$\|\mathbf{g}_i^{CAGT}\| = \|\mathbf{g}_i\|.$$

*Proof.* By definition, $\|\mathcal{R}(\mathbf{g}_i, \mathbf{g}_j)\| = \|\mathbf{g}_i\|$. Thus

$$\|\mathbf{g}_i^{\mathrm{CAGT}}\| \leq (1-\lambda)\|\mathbf{g}_i\| + \lambda\|\mathbf{g}_i\| = \|\mathbf{g}_i\|.$$

The two terms have nonnegative dot product, so equality holds. $\qquad\square$

## D.5 One-Step Improvement (Theorem D.5)

**Theorem D.5** (Single-Step Advantage). *Let $\theta^{MT}$ and $\theta^{CAGT}$ be obtained using $\mathbf{g}_1 + \mathbf{g}_2$ and $\mathbf{g}^{CAGT}$ with step size $t \leq 1/L$. If*

$$(\mathbf{g}_1 + \mathbf{g}_2)^\top \mathbf{g}^{CAGT} > \|\mathbf{g}_1 + \mathbf{g}_2\|^2,$$

*then for sufficiently small $t$, $\mathcal{L}(\theta^{CAGT}) < \mathcal{L}(\theta^{MT})$.*

*Proof.* By $L$-smoothness,

$$\mathcal{L}(\theta - t\mathbf{v}) \leq \mathcal{L}(\theta) - t(\mathbf{g}_1 + \mathbf{g}_2)^\top \mathbf{v} + \frac{Lt^2}{2}\|\mathbf{v}\|^2.$$

Choosing $\mathbf{v} = \mathbf{g}^{\mathrm{CAGT}}$ and sufficiently small $t$, the linear term dominates. $\qquad\square$

## D.6 Multi-Block and Multi-Task Extension

- For $K$ blocks, each block update is independent; the full gradient update is the sum over blocks.

- For $T > 2$, attention distributes across tasks; descent and conflict-reduction properties hold in expectation.

# E EXTENDED RESULTS

Table 3: Extended results for Rot-MNIST multi-task classification. Val. losses and accuracies are reported over 3 seeds with 3 reruns.

| Method | Loss (↓) | Digit Acc. (↑) | Rotation Acc. (↑) | Color Acc. (↑) |
|---|---|---|---|---|
| PCGrad | $0.2957 \pm 0.011$ | $92.14 \pm 0.02\%$ | $97.99 \pm 0.14\%$ | $99.97 \pm 0.04\%$ |
| GradNorm | $0.3645 \pm 0.015$ | $92.50 \pm 0.42\%$ | $96.55 \pm 0.39\%$ | $99.83 \pm 0.04\%$ |
| MGDA | $0.3025 \pm 0.012$ | $92.10 \pm 0.53\%$ | $98.00 \pm 0.26\%$ | $99.98 \pm 0.03\%$ |
| CAGrad | $0.2797 \pm 0.012$ | $91.96 \pm 1.07\%$ | $97.45 \pm 0.58\%$ | $99.99 \pm 0.01\%$ |
| CAGT (Ours) | $\mathbf{0.2525 \pm 0.021}$ | $\mathbf{93.57 \pm 0.58\%}$ | $\mathbf{98.11 \pm 0.54\%}$ | $99.99 \pm 0.02\%$ |

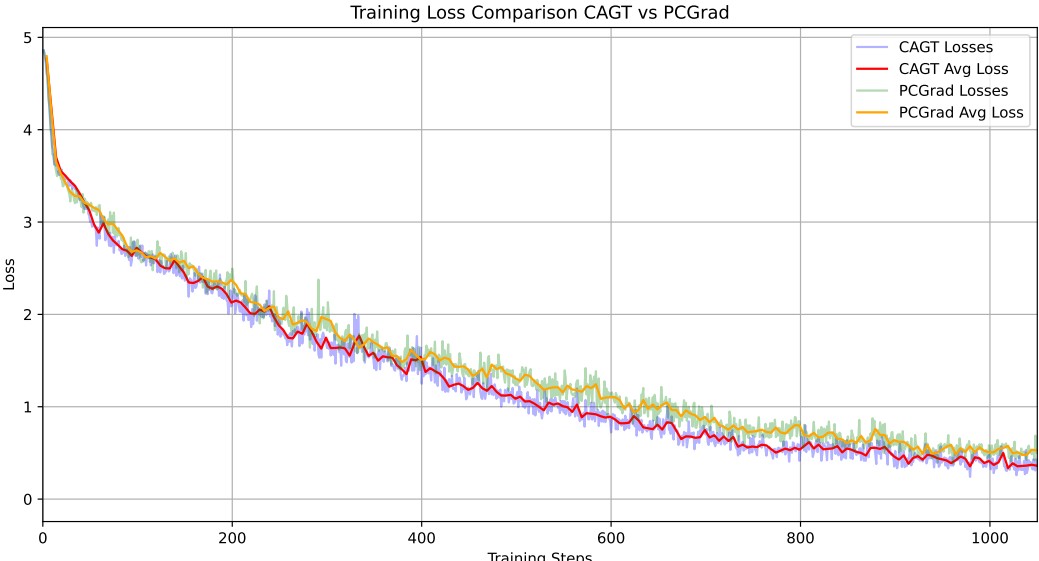

Figure 1: Train loss over epochs for CAGT vs. PCGrad for the ROT-MNIST Dataset

## E.1 ABLATION STUDY RESULTS

### E.1.1 TRAINING EFFICIENCY ANALYSIS

We analyze the number of training steps required for each method to reach a validation loss below 0.3 on Rot-MNIST. This metric provides insight into the practical convergence speed of CAGT compared to other multi-task optimization methods.

Table 4: Avg. Number of training steps to reach validation loss $< 0.3$ on Rot-MNIST. Only averages the cases where the model was able to achieve $< 0.3$ within 1200 training steps. Lower is better.

| Method | Avg. Training Steps to Loss $< 0.3$ |
|---|---|
| PCGrad | 1071.42 |
| GradNorm | ——- |
| MGDA | 1122.36 |
| CAGrad | 943.83 |
| CAGT (Ours) | **883.89** |

### E.1.2 ATTENTION MECHANISM ANALYSIS

Table 5: Comparison of attention similarity mechanisms under adaptive and fixed interpolation settings. Validation loss is reported as mean $\pm$ standard deviation over three runs and seeds on ROT-MNIST.

| Mechanism | Avg. Loss | |
| --- | --- | --- |
| | **Adaptive $\lambda$** | **Fixed $\lambda$** |
| Cosine | $0.2893 \pm 0.019$ | $0.2903 \pm 0.012$ |
| Cosine Squared | $\mathbf{0.2525 \pm 0.021}$ | $\mathbf{0.2643 \pm 0.013}$ |
| Euclidean | $0.3058 \pm 0.024$ | $0.3230 \pm 0.012$ |

Table 5 compares different attention similarity mechanisms under both adaptive and fixed interpolation settings. Across all configurations, cosine-squared similarity consistently outperforms standard cosine and Euclidean distance, achieving the lowest average validation loss and exhibiting reduced variance across runs.

Notably, adaptive interpolation improves performance for all mechanisms but the gain is most pronounced for cosine squared similarity. This suggests that sharper similarity contrast, amplified by squaring cosine similarity, interacts favorably with adaptive gradient mixing; likely by emphasizing high-confidence attention matches while suppressing noisy alignments. Euclidean distance performs worst overall, which indicates that magnitude sensitive similarity is less suitable in this setting.

Taken together, these results show that both the choice of similarity function and the use of adaptive interpolation materially affect CAGT performance. The combination of cosine-squared attention with adaptive $\lambda$ emerges as the most reliable configuration.

### E.1.3 TEMPERATURE AND INTERPOLATION COEFFICIENT ANALYSIS

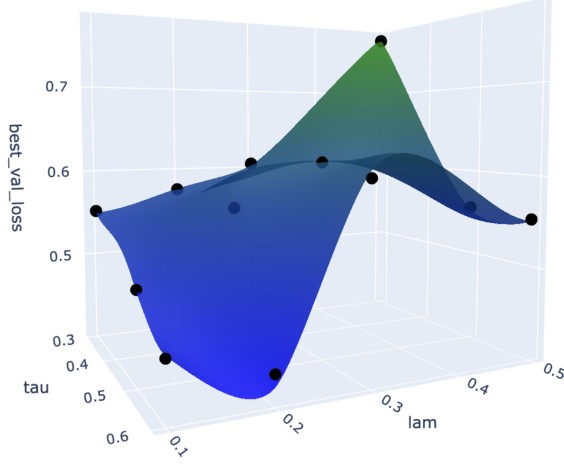

Figure 2: 3D Graph of parameter testing showing loss values by varying $\tau$ and $\lambda$

Figure 2 visualizes the validation loss surface obtained by sweeping the attention temperature $\tau$ and interpolation coefficient $\lambda$, with block size fixed at 8192. Each plotted point corresponds to an observed configuration, and the surface is produced via smooth interpolation constrained to pass through measured values.

From the sweep results, the lowest observed validation loss is achieved at $(\tau = 0.6, \lambda = 0.2)$, and it is followed closely by $(\tau = 0.6, \lambda = 0.1)$. By contrast, higher values of $\lambda$ consistently reduce

performance across all temperatures. The worst result occurs at $(\tau = 0.3, \lambda = 0.5)$. This indicates that overly aggressive gradient transplantation is detrimental when combined with low attention temperature. However, too low of a transplant interpolation value ($\lambda < 0.15$) diminishes the effects of CAGT and increases loss.

To better understand the trend, we treat the loss surface locally as a smooth function $f(\tau, \lambda)$ and approximate it using interpolation over the discrete measurements. First order partial derivatives $\partial f/\partial \tau$ and $\partial f/\partial \lambda$ are estimated numerically via finite differences between neighboring grid points. The sign change of these derivatives around ($\tau \approx 0.55$–$0.65$, $\lambda \approx 0.15$–$0.25$) and it suggests a local minimum region instead of a sharp optimum. Second order behavior inferred from curvature further supports that the optimum is broad.

# F   CAGT PREDICTIONS ON ROT-MNIST

Figure 3 shows CAGT predictions for digit classification, rotation estimation, and color assignment on ROT-MNIST. Each panel displays the input image transformed according to the predicted rotation and colored according to the predicted color class. The multi-task CAGT mechanism successfully captures all three tasks simultaneously.

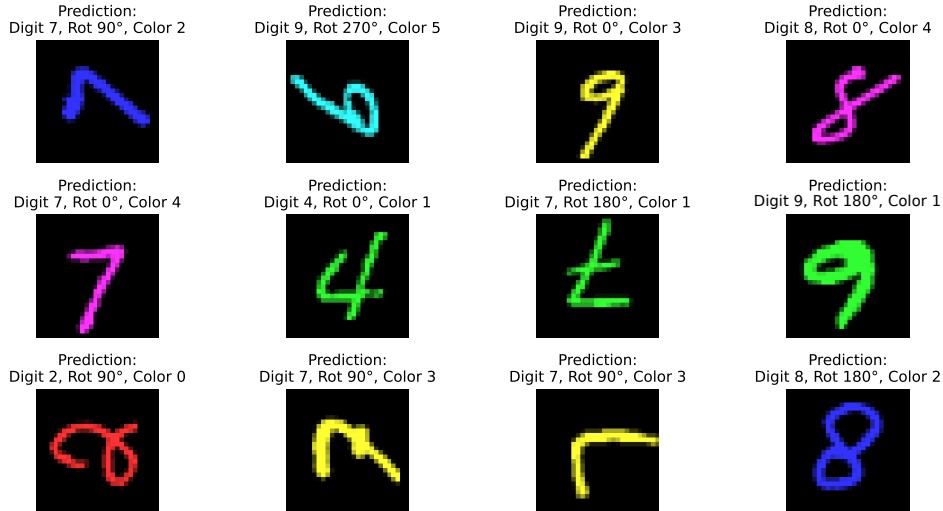

Figure 3: CAGT predictions for digit, rotation, and color on ROT-MNIST (12 random images pulled). The labels indicate the predicted digit, rotation (in degrees), and color class. The color indices correspond to the following mapping: 0 = red, 1 = green, 2 = blue, 3 = yellow, 4 = magenta, 5 = cyan.

