# OpenReview forum: "Cross-Attention Gradient Transplantation (CAGT): Mitigating Gradient Conflict in Multi-Task Deep Learning Through Cross-Blockwise Attention"
_ICLR.cc/2026/Workshop/Sci4DL — Submitted to Sci4DL 2026_

### Official Review · Reviewer_HHPD · 2026-02-26

**Fit:** 3
**Significance:** 2
**Confidence:** 3

**Summary:**

The paper proposes Cross-Attention Gradient Transplantation (CAGT) to mitigate gradient conflict in multi-task learning. The method partitions parameters into fixed blocks, detects conflicts at the block level via cosine similarity, and, for conflicting blocks, reconstructs an alternative update direction via cross-task attention (a cosine-similarity-based convex combination), followed by magnitude rescaling and interpolation with the original gradient. Empirically, CAGT outperforms baselines such as PCGrad and CAGrad on Rotated MNIST and CelebA.

**Strengths:**

1. The paper is clearly written and easy to follow, with a motivation around localized gradient conflict and a step-by-step description of the procedure.

2. The decision to resolve conflict blockwise (treating each block as an atomic unit for alignment and transformation) is interesting and plausibly better matched to the observation that conflicts can be localized rather than global.

**Suggestions:**

1. As CAGT’s reconstruction step forms a convex combination of other-task gradients within a conflicting block, it would strengthen the paper to include additional baselines that isolate whether the gains come from attention vs. “just mixing”:

              a.) Uniform average baseline: for conflicting blocks, replace the attention weights with uniform weights over the negatively-aligned task blocks.
              b.) Random convex mixing baseline: sample random weights on the simplex and report mean/variance.


2. The algorithm explicitly takes block size $B$  as an input hyperparameter. However, it is not fully clear how $B$ is chosen in practice for the presented models, or what “block” corresponds to (e.g., contiguous flattened chunks vs. parameter-tensor-aware grouping). The authors should add an experiment varying block size from very small (approaching “each parameter is its own block”) to large (approaching “whole gradient is one block”), and report both performance and overhead.

3. The computation overhead is missing from the paper; the authors should discuss briefly what the computation overhead of the proposed method is.

---

### Official Review · Reviewer_rqU2 · 2026-02-26

**Fit:** 2
**Significance:** 2
**Confidence:** 2

**Summary:**

The authors propose cross-attention gradient transplantation, a scheme for improving the training of models on multiple tasks at once. The method breaks model parameters into blocks, and adjusts gradients to ease multi-task training by using the attention weights across parameter blocks.

**Strengths:**

The paper is clear and easy to read, and the work is interesting and I believe worth pursuing further. I recommend acceptance to the workshop, and hope the authors expand the work further to investigate some key questions (see below).

**Suggestions:**

I am interested in how the blocking procedure affects results -- how does performance chance as the blocking scheme changes? Does the optimal blocking scheme scale with the number of tasks, or some measure of task similarity?

I am not sure if this is possible, but I wonder if one could extract a task similarity measure from the cross-block attention weights?

I think it would also be important to ensure the method performs well in a setting with many tasks, as often occurs in LLM training.

I believe experiments along these lines would be quite interesting, and would also improve the paper's fit with the workshop, as currently there is not much analysis of *why* such a scheme improves multi-task performance. Nonetheless, I believe the authors have identified a promising setting to investigate many of these scientific questions.

---

### Meta-Review · Area_Chair_KHM8 · 2026-02-28

**Recommendation:** Reject

**Metareview:**

This appears to be a legitimate paper on network optimization, and it got positive reviews. However, it is primarily applied optimization, and does not study the *science* of deep learning. Proposed scientific insights are concluded post-hoc from optimization gains with little direct evidence and not investigated separately or directly. This is a fine paper, but it is not in scope for this workshop.

---

### Decision · Program_Chairs · 2026-03-02

Reject